# YOLOv7-Plum: Advancing Plum Fruit Detection in Natural Environments with Deep Learning

**DOI:** 10.3390/plants12152883

**Published:** 2023-08-07

**Authors:** Rong Tang, Yujie Lei, Beisiqi Luo, Junbo Zhang, Jiong Mu

**Affiliations:** 1College of Information Engineering, Sichuan Agricultural University, Ya’an 625000, China; 2022319023@stu.sicau.edu.cn (R.T.); 2021319016@stu.sicau.edu.cn (B.L.); 2021219004@stu.sicau.edu.cn (J.Z.); 2Sichuan Key Laboratory of Agricultural Information Engineering, Ya’an 625000, China; 3College of Information and Electrical Engineering, China Agricultural University, Beijing 100193, China; s20233081800@cau.edu.cn

**Keywords:** plum, deep learning, computer vision, smart agriculture, object detection

## Abstract

The plum is a kind of delicious and common fruit with high edible value and nutritional value. The accurate and effective detection of plum fruit is the key to fruit number counting and pest and disease early warning. However, the actual plum orchard environment is complex, and the detection of plum fruits has many problems, such as leaf shading and fruit overlapping. The traditional method of manually estimating the number of fruits and the presence of pests and diseases used in the plum growing industry has disadvantages, such as low efficiency, a high cost, and low accuracy. To detect plum fruits quickly and accurately in a complex orchard environment, this paper proposes an efficient plum fruit detection model based on an improved You Only Look Once version 7(YOLOv7). First, different devices were used to capture high-resolution images of plum fruits growing under natural conditions in a plum orchard in Gulin County, Sichuan Province, and a dataset for plum fruit detection was formed after the manual screening, data enhancement, and annotation. Based on the dataset, this paper chose YOLOv7 as the base model, introduced the Convolutional Block Attention Module (CBAM) attention mechanism in YOLOv7, used Cross Stage Partial Spatial Pyramid Pooling–Fast (CSPSPPF) instead of Cross Stage Partial Spatial Pyramid Pooling(CSPSPP) in the network, and used bilinear interpolation to replace the nearest neighbor interpolation in the original network upsampling module to form the improved target detection algorithm YOLOv7-plum. The tested YOLOv7-plum model achieved an average precision (AP) value of 94.91%, which was a 2.03% improvement compared to the YOLOv7 model. In order to verify the effectiveness of the YOLOv7-plum algorithm, this paper evaluated the performance of the algorithm through ablation experiments, statistical analysis, etc. The experimental results showed that the method proposed in this study could better achieve plum fruit detection in complex backgrounds, which helped to promote the development of intelligent cultivation in the plum industry.

## 1. Introduction

The plum, a woody plant of the genus Plum in the family Rosaceae, is native to southeastern China and is one of the oldest cultivated fruits in the world, and it is now cultivated in most areas around and south of the Qinling Mountains in China. China is the largest grower and producer of plums, and its plum industry has an important position in both domestic and international economic markets [1]. From 2010–2022, China’s plum production increased from 5,521,700 tons to 6,626,300 tons, accounting for 51.6% to 55.2% of global plum production, and the harvested area increased from 1665.2 thousand hectares to 1946.5 thousand hectares; the share of the global plum harvested area increased from 69.1% to 74.8% (Food and Agriculture Organization of the United Nations, 2022). The domestic plum supply is abundant, but the overall import volume is still greater than the export volume due to the huge consumer market [2]. Plum fruits are about 4 cm in diameter, with a single fruit weight of about 50 g to 210 g. The fruit is full, delicious, and juicy, and the flesh contains basic components, such as protein, carbohydrates, and fat, as well as antioxidant-active substances, such as flavonoids, carotenoids, and anthocyanins, which are rich in nutritional value, and the plum is one of the main excellent fresh fruits in summer [3,4]. In addition, the fruit is resistant to storage and transportation, which is favored by many operators and consumers [5].

During the growth of plum trees, too much fruit at the same time will consume a lot of nutrients, weaken the tree’s vigor, and affect the quality and size of the fruit, so it is necessary to artificially reduce the number of fruits. However, if plum trees produce less fruit, it will affect economic performance and require consideration of changing varieties [6,7]. At the same time, changes in climate, soil, and the surrounding environment affect the growth and development of fruit trees, with pests and diseases causing greater economic losses [8]. Therefore, continuous testing and analysis are needed during the growth of plums, thus avoiding problems such as yield reduction due to fruit yield and pests and diseases. However, the small size of plum fruits and the complexity of the actual plum orchard environment make the detection of plum fruits difficult due to leaf shielding and fruit overlap. Most of the traditional plum farming industry uses manual visual estimation to determine the number of fruits and the problems of pests and diseases [9]. In recent years, there has been an unprecedented increase in labor costs due to the continuous loss of agricultural labor, and this problem has become more prominent after the COVID-19 pandemic [10]. In summary, the design of an efficient and accurate method for plum fruit detection is necessary to estimate the number of fruiting plum trees and to facilitate research on pest and disease identification. 

With the continuous development of computer science and technology, computer vision technology in deep learning has penetrated people’s production life, and its importance is becoming more and more prominent [11]. More and more researchers had put computer vision technology into agricultural production and have a wide range of practical applications [12,13], such as intelligent greenhouses [14], agricultural robots [15], etc. Target detection is an important task in the field of computer vision, and its main job is to identify and localize the target of interest in the input image, which solves the problem of where and what the object is in the image [16]. Target detection techniques have achieved good results in agricultural production areas, such as pest and disease monitoring [17,18], crop yield prediction [19], and crop growth monitoring [20]. The applications of target detection models for crop fruit detection [21] can be divided into two categories: detection using one-stage models and detection using two-stage models [22]. In the detection process, two-stage models separate the proposed region from the background and then classify and localize the target [23], among which the representative one uses the region-CNN (RCNN) series for detection. Sa et al. [24] used the improved Faster-RCNN [25] based on multi-modal (RGB and NIR) information fusion to detect a variety of fruits, including apples, bell pepper, and melon, with an average detection accuracy of 0.838. Yu et al. [26] detected strawberry fruits based on Mask-RCNN with a detection accuracy of 0.898. Fu et al. [27] built an algorithm consisting of ZFNet and Faster R-CNN architecture of VGG16 to detect apples in vertical fruit wall trees, and the fruit detection accuracy was improved by 2.5% after removing the background tree with a depth filter. RCNN series have a high detection accuracy for crop fruit detection, but the detection speed is relatively slow due to the large number of calculation parameters and complex models, which posed certain challenges for computing hardware in practical applications. The one-stage model uses an initial anchor frame to locate the target region and directly predict the target class [28]. The representative of one-stage models is the You Only Look Once (YOLO) series, and deep learning models based on YOLO are widely used for fruit detection and recognition [29]. Fu et al. [30] cut the less important network layer by analyzing the characteristics and network structure of bananas. As a result, the YOLOv4 model was improved to form the YOLO-Banana model, and the average accuracy (AP) of banana string and banana stem was 98.4% and 85.98%, respectively. Li et al. [31] improved the YOLOX model by modifying the loss function and adding attention to detect sweet cherries and achieved a detection accuracy of 84.96%, an improvement of 2.34% relative to the initial YOLOX model. Wang and He [32] detected young apple fruits based on the YOLOv5 model with channel pruning, and the detection accuracy reached 95.80% with an average detection time of 8 milliseconds per image. Compared with the RCNN series, the YOLO series achieved similar performance and higher computational efficiency.

Based on the above studies, the YOLO model has great potential for fruit detection under natural conditions. The YOLOv7 model additionally employs efficient aggregation networks, reparameterized convolution, positive and negative sample matching strategies, auxiliary head training, and model scaling [33,34], which enables the model to significantly improve the feature extraction capability of the target. In the range of 5 FPS to 160 FPS, the YOLOv7 model is faster and more accurate than any known target detector [35]. This paper selected the YOLOv7 network as the initial detection model after taking into account the object to be detected, the detection accuracy of the network, and the need for lightweighting.

The spatial Pyramid Pooling (SPP) module [36] was proposed by Kai-Ming He in 2015; its main role is to solve the problem of uneven input image size. The SPP structure used in YOLOv7 is Cross Stage Partial Spatial Pyramid Pooling (CSPSPP), although its effect is better than SPP and Spatial Pyramid Pooling-Fast (SPPF) used in YOLOv5, the feature extraction speed of CSPSPP is slowed down due to the increasing number of parameters and calculation amount. Upsampling refers to inserting new elements between pixel points based on the original image pixels by using a suitable interpolation algorithm, i.e., enlarging the original image to make it higher resolution [37]. The interpolation algorithm used in YOLOv7 for upsampling is nearest interpolation, which inserts the pixel values as its own by finding the pixel points closest to the location of the inserted pixel points, which is less computationally intensive and simpler [38]. However, it only uses the gray value of the pixel closest to the sampling point to be measured as the gray value of that sampling point without considering the influence of other neighboring pixel points; thus, the gray value has obvious discontinuity after resampling, and the image quality loss is large, which will produce obvious mosaic and jagged phenomenon, so determining how to choose a more suitable interpolation algorithm in the upsampling process is worth studying. In computer vision, the method of focusing attention on important regions of an image and discarding irrelevant ones is called the attention mechanism [39]. Attention mechanisms are a common data processing method that applies the human perception style and the behavior of attention to the machine so that the machine learns to perceive the important and unimportant parts of the data [40]. The introduction of an attention mechanism can make the network focus on the target area, enhance the effective feature extraction ability of the network, reduce the interference of invalid targets, and obtain more detailed information [41,42]. The Convolutional Block Attention Module (CBAM) is a simple and effective precursor convolutional attention module proposed by Woo et al. [43]. It focuses on both spatial focus and channel focus and can be applied in YOLOv7 networks to improve the feature perception of the network for targets in images [44,45,46].

In view of the above limitations, the aim of this study is to achieve the fast and accurate detection of plum fruits in natural environments, so this paper proposed a YOLOv7-based target detection algorithm, YOLOv7-plum. By modifying the structure of the convolutional layer, reducing the number of parameters in the process of feature extraction, and changing the parallel structure to a more lightweight series structure, this paper improved the CSPSPP structure used in YOLOv7 to Cross Stage Partial Spatial Pyramid Pooling–Fast (CSPSPPF) structure. To further improve the upsampling operation in YOLOv7, the nearest interpolation in the original model was replaced with bilinear interpolation [47]. In addition, two CBAMs were embedded in the network structure of YOLOv7 to improve the perception of the model for different spaces and different channels so that the model could focus more on the key information in the image. Finally, the feasibility and reliability of the method were verified in this paper with ablation experiments and a statistical analysis. The experimental results showed that our proposed YOLOv7-plum algorithm could better achieve plum fruit detection in complex backgrounds, which helped to promote the estimation of fruit number of plum trees and yield prediction of plum and laid the foundation for the research of a plum pest detection model.

The remainder of this paper is organized as follows. Section 2 introduces the details of the dataset used in this study and the plum detection algorithm model YOLOv7-plum. Section 3 evaluates the performance of the YOLOv7-plum algorithm through experiments. Section 4 shows the discussion results. Finally, Section 5 summarizes the work of this study.

## 2. Result

### 2.1. Impact of Data Enhancement

In order to investigate the effect of data enhancement on the experimental results, this paper used the unenhanced dataset (Name dataset1) and the enhanced dataset (Name dataset2) to conduct experiments on YOLOv7 and YOLOv7-plum, respectively, with the datasets divided in the ratio of 8:1:1 and the epoch set to 100. The experimental results are shown in Figure 1 and Table 1.

It can be seen that compared with the experimental results on dataset 1, the relevant indexes of both YOLOv7 and YOLOv7-plum in dataset 2 improved by 2% to 4%. The experimental results demonstrated that the data enhancement method used in this paper effectively improved the model detection accuracy.

### 2.2. Comparison with YOLOv7

According to Table 1, the AP of YOLOv7-plum is better than YOLOv7 under both datasets; the AP of YOLOv7-plum is 94.91%, and recall is 93.26% on dataset 2, which is 2.03% and 0.87% higher than YOLOv7, respectively. The model size is only 0.1 MB larger. This paper randomly selected images that were not used in model training and used YOLOv7-plum and YOLOv7, respectively, for prediction. Their pairs are shown in Figure 2.

According to Figure 2, the plum fruits at the red box position in Figure 2a are not detected by YOLOv7, and in Figure 2b, YOLOv7-plum detects the fruits at these two positions. This indicates that YOLOv7 is prone to miss detection in the face of occlusion, and YOLOv7-plum alleviates the missed detection to a certain extent. In contrast, plums in the natural environment cannot avoid the shielded situation, so YOLOv7-plum is considered to be more suitable for plum detection in the natural environment.

In summary, from the comprehensive performance of the model, the YOLOv7-plum model had higher detection accuracy and the size of the model had not increased significantly, and it could perform the plum fruit detection task efficiently and accurately and can be run on lightweight devices.

### 2.3. Ablation Experiments

In order to verify the effectiveness of the modules added in this paper, ablation experiments were designed based on the data set constructed in this paper, with S representing the CSPSPPF module, C representing the CBAM, and B representing the bilinear interpolation module. The results of the ablation experiments are shown in Table 2. In Table 2, “√” indicates that the improved method of the column is used, and “×” indicates that this improvement method is not used.

From Table 2, YOLOv7-S is almost the same as YOLOv7 in terms of various indexes because compared with CSPSPP, CSPSPPF only reduced the amount of computation and had no significant impact on the model effect. Therefore, this paper will not discuss the influence of only adding the CSPSPPF module on the model detection (that is, the comparison between YOLOv7-C and YOLOv7-SC, YOLOv7-B and YOLOv7-SB, and YOLOv7-CB and YOLOv7-SCB is no longer performed.); only the experiments with SPPFCSPC (i.e., YOLOv7-SC, YOLOv7-SB, and YOLOv7-SCB) are analyzed.

Both YOLOv7-SC and YOLOv7-SB improved AP to some extent, but YOLOv7-SC had the problem of lower precision and YOLOv7-SB had the problem of lower recall. For YOLOv7-SCB, it improved AP by 2.03% while the rest of the evaluation metrics were similar to or better than the original model, which proved the effectiveness of the various modules in our approach.

### 2.4. Comparison with Mainstream Networks

On dataset 2, experiments were conducted on several mainstream target detection networks as well as the YOLOv7-plum proposed in this paper, and the results of the experiments are shown in Table 3. In terms of AP metrics, The AP of YOLOv7-plum is 94.91%, which is significantly better than 90.83% of faster R-CNN, 92.68% of YOLOv5, and 93.94% of DETR. Since the AP of faster R-CNN is the lowest and its model size is larger, faster R-CNN is not discussed anymore. Although YOLOv5s has the smallest model size, the AP is 2.23% lower than YOLOv7-plum, and the model size of YOLOv7-plum is acceptable compared to the reduced AP. The AP of DETR is the closest to that of YOLOv7-plum, only 0.97% lower than that of YOLOv7-plum, but the model size is about 6.6 times that of YOLOv7-plum. The AP of YOLOv8l is 1.01% lower than YOLOv7, and the model size increases by 12.2 MB, which is still inferior to YOLOv7-plum. The above results showed that YOLOv7-plum had higher detection accuracy in plum fruit detection, which was better than the current mainstream target detection networks and can support continuous detection and analysis during plum growth.

### 2.5. Statistical Analysis

To verify the robustness of the model, new plum images were acquired through growers in Gulin County, which were taken on 22 October 2022, as the plums are not yet fruiting now. This is called as dataset 3. It contained a total of 81 images, which were classified into four categories, A, B, C, and D, according to the number of plums in the images, and each category was described as follows:

A: The number of plums in a single picture was less than or equal to 5;

B: The number of plums in a single picture was greater than 5 and less than or equal to 10;

C: The number of plums in a single picture was greater than 10 and less than or equal to 15;

D: The number of plums in a single picture was greater than 15.

Eighty-one images were manually counted in dataset 3, which had a total of 504 plum fruits. This paper utilized YOLOv7-plum and YOLOv7 for detection and counting respectively on dataset 3. Examples of the results of detection and counting by YOLOv7-plum are shown in Figure 3. Figure 3 shows the detection of YOLOv7-plum on four types of images, A, B, C, and D, respectively, and shows the counting results in the upper left corner of the images. As a result, YOLOv7 detected 418 plum fruits, and YOLOv7-plum detected 483 plum fruits. The counting results of YOLOv7-plum were closer to the real number of fruits, which indicated that our proposed method was superior to YOLOv7.

A statistical analysis was conducted on the detection results of the two models, which are clearly shown in Figure 4. The linear regression equation (red line) of the YOLOv7-plum counting results fits better with the true value (green line). The correlation index R^2^ of YOLOv7-plum is 0.98, which is 0.09 higher than the correlation coefficient R^2^ of YOLOv7, and the mean absolute percentage error (MAPE) of YOLOv7-plum is 2.72%, which is 11.53% lower than the MAPE of YOLOv7. This proves that the detection error of YOLOv7-plum is smaller than that of YOLOv7.

## 3. Materials and Methods

### 3.1. Dataset

#### 3.1.1. Experiment Field and Data Acquisition

Gulin County, Luzhou City, Sichuan Province, with fertile land and abundant light, has about 35,000 mu of plum fruit groves and a perennial production of about 15 million kg, making it one of the best places for high-quality plums in China (People’s Government of Gulin County, 2022). In recent years, the Gulin County government and departments at all levels have promoted the development of the local plum industry by establishing professional cooperatives and combining modern agriculture and tourism industries, and the plum industry has now become a characteristic and advantageous industry in Gulin County. The data collection area of this study is a plum orchard located in Gulin County with GPS coordinates of 105°96′ E and 28°10′ N. Figure 5 shows the geographic location.

The dataset was captured in July 2022, and the plums were all grown in their natural state. To meet sample diversity and improve network robustness, plum images were acquired by multiple devices, including a Redmi K20 pro (Xiaomi Technology Co., Ltd., Beijing, China), a Canon camera (60D, 18 megapixels, 18–200 IS lens), and a Sony camera (Alpha 6400, 24.2 megapixels, 17–70 mm lens). Figure 6 shows sample images taken with both devices. We took 2624 images of plums, and after we manually filtered them to remove some duplicates and similar data, 700 images were finally selected as our initial data set.

#### 3.1.2. Data Enhancement and Data Annotation

In order to increase the diversity of the training set and improve the generalization ability [48] of the model in this study, 700 images of the initial dataset were data enhanced by flipping them at random angles, mirror transforming them, adjusting their brightness, adding random noise, etc. An example of data enhancement is shown in Figure 7, Figure 7a shows the original images, and Figure 7b shows the data enhanced images. Finally, an experimental dataset of 1220 images was obtained, with each image containing anywhere from 1 to 20 plum fruits. 

According to the proportion of 8:1:1, the dataset was divided into a training set, a validation set, and a test set, and the dataset was labeled using the open-source script LabelImg from https://github.com/HumanSignal/labelImg (accessed on 9 July 2023). An example of the annotation of the dataset is shown in Figure 8. The labeling process produced a total of 8745 labeled boxes, of which the training set contained 7042 labeled boxes, the validation set contained 834 labeled boxes, and the test set contained 869 labeled boxes. The division of the dataset and the distribution of the number of labeled boxes are shown in Table 4.

### 3.2. Experimental Platform

This study was carried out on a professional server. The frame image source was PyTorch 1.8.1, the training environment of Python 3.8 (ubuntu18.04) was used, and Cuda 11.1 was used as the computing architecture. The hardware GPU was RTX 3080 × 1, and the video memory was 10 GB. The CPU adopted a 15-core Inter (R) Xeon(R)Platinum 8358P CPU @2.60GHz, and the memory was 80 GB.

### 3.3. Deep Learning Network Construction

#### 3.3.1. CSPSPPF Module

The Spatial Pyramid Pooling (SPP) module [36] can convert feature maps of arbitrary size into fixed-size feature vectors in the feature extraction process. Glenn Jocher, one of the authors of YOLOv5, has adjusted the structure of SPP to propose the SPP-Fast(SPPF) module [49], which makes the model much less computationally intensive and improved the model speed. The SPP module used in YOLOv7 is Cross Stage Partial Spatial Pyramid Pooling (CSPSPP) [33]. CSPSPP introduces a CSP structure based on the traditional SPP structure, which performs better than SPPF on the COCO dataset, but the number of parameters and calculation amount are greatly improved. 

The idea of the improvement of SPPF over SPP was referenced. In the CSPSPP module used in YOLOv7, two 5 × 5 convolution operations replaced a 9 × 9 convolution operation and three 5 × 5 convolution operations replaced a 13 × 13 convolution operation, and the parallel structure was changed into a more lightweight series structure, which reduced the amount of computation while ensuring the same perceptual field and improved the computational efficiency of the model. The improved module was called Cross Stage Partial Spatial Pyramid Pooling–Fast (CSPSPPF). The structure diagram of SPP, SPPF, CSPSPP, and CSPSPPF is shown in Figure 9.

#### 3.3.2. Bilinear Interpolation

The interpolation algorithm is a basic and important task in image scaling processing to fill the gaps between pixels during image transformation. Common interpolation algorithms include nearest neighbor interpolation [38], bilinear interpolation [47], and bicubic interpolation [50]. The YOLOv7 model uses the nearest neighbor interpolation algorithm in the upsampling process.

The nearest neighbor interpolation finds the nearest pixel from the location of the inserted pixel and inserts it as its own pixel value; bilinear interpolation calculates the pixel value of the inserted pixel location based on the pixel values of the four points closest to the inserted pixel location; bicubic interpolation obtains the inserted pixel point pixel value by a weighted average of the 16-pixel points closest to the inserted pixel location in the rectangular grid. Experiments were conducted to transform the data in the model run by twice the number of pixels using three interpolation algorithms, derived the transformed images using different interpolation algorithms, and calculated the time spent on transforming the images. Examples of the results of transforming the data in this study using the three interpolation algorithms are shown in Figure 10, and the time spent on transforming the images is shown in Table 5. 

From Figure 10 and Table 5, it can be seen that among these three interpolations, bicubic interpolation worked better than bilinear interpolation and nearest neighbor interpolation but was slower than nearest neighbor interpolation and bilinear interpolation in terms of computational speed. The quality of the transformed image after bilinear interpolation was between nearest neighbor interpolation and bicubic interpolation, but in terms of execution time, bicubic interpolation increased the time spent by 77.75% compared to nearest neighbor interpolation, and bilinear interpolation increased the time spent by only 22.21% compared to nearest neighbor interpolation. Considering the time feasibility and the quality of the transformed image, bilinear interpolation was used instead of the nearest neighbor interpolation used in YOLOv7 in order to obtain better target detection results while keeping the speed as fast as possible.

#### 3.3.3. Attention Mechanism

The introduction of an attention mechanism in deep learning can make the network focus on the target region, reduce the interference of invalid targets and improve the effective feature extraction ability of the network, thus improving the detection of targets of concern. The Convolutional Block Attention Module (CBAM) is a lightweight attention module that focuses on both channel and space dimensions [43], and the module structure is shown in Figure 11.

Two CBAMs were embedded between the head and backbone of the YOLOv7 network structure to improve the performance of the network. After embedding the CBAMs the network was able to serially generate attention feature map information in both channel and spatial dimensions separately during feature extraction and then carry out adaptive feature correction between the two kinds of information and the original input feature map to generate the final feature map. YOLOv7 with the addition of CBAM increased the channel weights of the detection target and expanded the perceptual field of the original image. Therefore, in this study, the key information of plum fruit can be better focused from the complex background. 

#### 3.3.4. YOLOv7-Plum Deep Learning Network Structure

YOLO (You Only Look Once) is a typical single-stage target detection algorithm. YOLOv7 is the structure introduced in the YOLO series in 2022, and it offers significant improvements in feature extraction for targets compared to previous models [33]. The basic architecture of the YOLOv7 consists of three modules: input, trunk, and detection head. The input module first preprocesses the image by aligning it to an image size of 640 × 640. The backbone module consists of three parts, CBS, ELAN, and MP. The input image is passed through the backbone module for feature extraction. YOLOv7 has three detection heads; after feature fusion, the detection heads will pass the feature information to the output layer, which predicts the location and class of the target and generates the corresponding bounding box [51,52].

Two CBAMs were added between the head and backbone in the YOLOv7 network structure to improve the model’s ability to perceive different spaces and different channels, enabling the model to pay more attention to the plum fruit as the focal information in the complex background. Then, the nearest neighbor interpolation used in the up-sampling of the original network was changed to the more effective bilinear interpolation, which resulted in a significant improvement of the image quality during the up-sampling process without a significant increase in the training time of the model. Finally, the improved CSPSPPF structure replaced the CSPSPP structure in the original network, which reduced the number of parameters and the computational effort and improved the computational efficiency of the model while ensuring that the sensory field of the model remained unchanged. The improved network achieved better results for plum fruit detection in complex natural environments, and the improved network was called YOLOv7-plum. The network structure of YOLOv7-plum is shown in Figure 12.

### 3.4. Evaluation Indicators

In this paper, precision, recall, and average precision (AP), the most commonly used evaluation indexes in object detection, were used as the evaluation indexes in this experiment.

According to the combination of the real category and prediction category, the sample was divided into four cases: true positive (TP), false positive (FP), true negative (TN), and false negative (FN). The parameters were defined in Table 6.

Precision can be understood as “In the data that the model predicts to be correct, its true value is the proportion of the samples that are correct”. The formula was:(1)$$Precision=\frac{TP}{TP+FP}$$

Recall is “The percentage of samples that are predicted to be correct by the model out of all samples that are correct”. The formula was:(2)$$Recall=\frac{TP}{TP+FN}$$

In the PR (precision–recall) curve, P stood for precision and R stood for recall, and it stood for the relationship between precision and recall. Generally, recall is set as the abscissa and precision as the ordinate.

AP meant the area under the PR curve, and AP was also an indicator of PR. The better the classifier, the higher the AP value.

## 4. Discussion

### 4.1. Feasibility Analysis

In order to promote the development of intelligent cultivation of the plum industry, this study applied the YOLOv7 model to the fruit detection of plums in the natural environment and obtained the YOLOv7-plum plum fruit detection model by improving the limitations of YOLOv7. The results showed that the YOLOv7-plum model could perform the task of plum fruit detection better. For the feasibility of the proposed method in this paper, we developed the following discussion:

(1) The model size of our proposed YOLOv7-plum network was 71.4 MB, which was almost the same as that of YOLOv7, and it took only 0.0193 s to infer a single image in the experimental environment (GPU is RTX 3080 × 1 with 10 GB of video memory). Our model inferred a single image time of 0.186 s in a personal laptop environment (GPU of NVIDIA GeForce GTX 1050 × 1 with 2 GB of video memory). Then, we exported the model and deployed it on a development board environment (ARM^®^ Cortex^®^-A53, quad-core, 1.2 G) for testing, and the inference time for a single image was 0.842 s. The experimental results showed that our model could meet the requirements of hardware deployment and the processing speed could meet the practical needs;

(2) Few studies have been conducted for intelligent plum planting, and the models used include YOLOv4 [53], faster RCNN, and EfficientDet [4], with AP values ranging from 59.81% to 88.56%. The models of the above studies are not novel enough, and the detection accuracy is low. This study promoted the application of novel computer vision algorithms on plum fruit detection to some extent;

(3) YOLOv7-plum has low equipment requirements and can be deployed on platforms with limited computing power, providing great convenience for the practical application of the model.

In summary, the method proposed in this study could meet practical needs and had certain advantages over existing studies, which could well accomplish the task of plum fruit detection in the natural environment and was expected to be further used for tasks such as robotic picking and plum pest detection.

### 4.2. Contribution to Intelligent Cultivation of the Plum Industry

At present, most plum-growing industries still use traditional manual techniques, which have disadvantages, such as low efficiency, high labor cost, and subjectivity, so it is necessary to carry out research on the intelligent growth of the plum industry. This study achieved the detection of plum fruits in a natural environment using a deep learning algorithm, and the contribution to the intelligent planting of plums was as follows:

(1) Manual picking of plum fruits is a labor-intensive task that takes a lot of labor cost, while robotic picking can help reduce labor requirements and can improve efficiency. This study achieved real-time detection of plum fruits, which could be further used for robotic picking [27];

(2) Due to the suddenness of crop pests and diseases, real-time monitoring of plum growth was beneficial to the timely detection of pests and diseases. This study can be used as a basis for the study of plum fruit pests and disease detection in the natural environment;

(3) Manual judgment of whether plum trees need to be thinned was influenced by human subjectivity, so this study can realize the counting of young fruits with fruit detection on plum fruit trees, which can avoid subjective influence and help growers optimize orchard management [32]. It is also possible to use a combination of two-sided imaging and four-sided imaging for plum yield prediction based on this study [54].

## 5. Conclusions

In this study, deep learning technology was applied to the detection of plum fruits, and the YOLOv7 model was improved to obtain a YOLOv7-plum model with a better detection effect on plum fruits. First, the collected plum image data were enhanced and marked, and then, the plum fruit detection data set was formed. Second, the CSPSPP structure used by YOLOv7 was modified to the CSPSPPF structure to improve the computational efficiency in the feature extraction process. Third, bilinear interpolation was used to replace the nearest neighbor interpolation used by the original model to improve the image quality in the upsampling process. Finally, two CBAMs were embedded between the head and backbone of the model structure to make the model pay more attention to the important information of plum fruit in the image during operation. The main conclusions were as follows:

(1) The AP of the YOLOv7-plum model proposed in this study reached 94.91%, 2.03% higher than that of the YOLOv7 model. It was found that the YOLOv7-plum model could alleviate the missing detection of the YOLOv7 model to some extent. Therefore, the improved method is believed to be effective, and the YOLOv7-plum model can effectively support continuous detection and analysis during plum growth;

(2) Through comparative tests, it was found that the detection accuracy of the YOLOv7-plum model was better than that of the current mainstream target detection network model. The size of 71.4 M was acceptable, and only 0.0193 s was needed to process a single image. The results showed that YOLOv7-plum could accurately complete the plum fruit detection task and could be run on lightweight devices;

(3) The method proposed in this study could effectively realize the detection of plum fruit under the natural background and reduced the problems of traditional manual detection, such as low efficiency, high cost, and subjectivity, which is conducive to the development of intelligent planting in the plum industry, such as yield prediction and pest detection.

## Figures and Tables

**Figure 1 plants-12-02883-f001:**
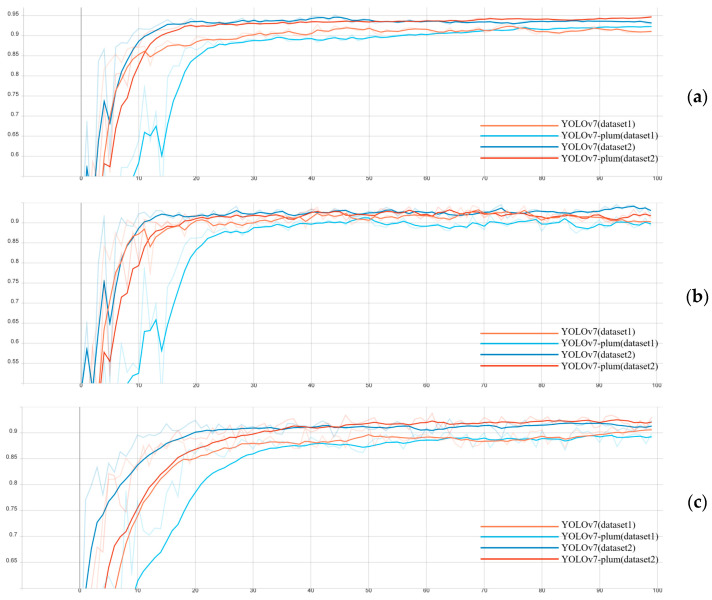
Four model indicator charts. (**a**) AP; (**b**) Precision; (**c**) Recall.

**Figure 2 plants-12-02883-f002:**
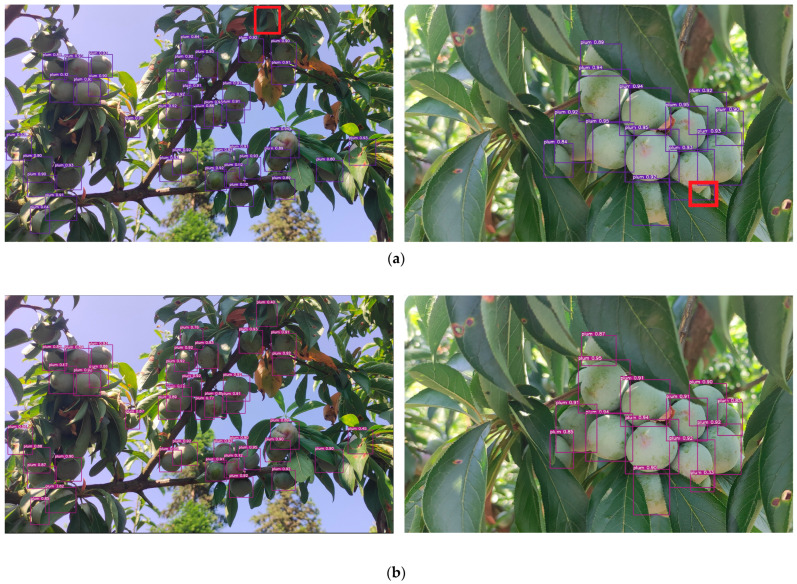
Comparison of detection of YOLOv7-plum and YOLOv7. (**a**) The detection results of YOLOv7; (**b**) The detection results of YOLOv7-plum.

**Figure 3 plants-12-02883-f003:**
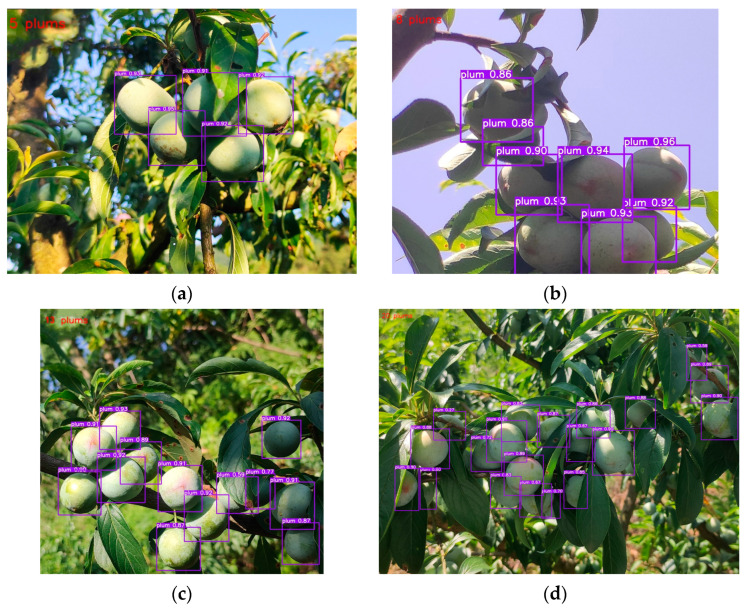
Examples of the results of detection and counting by YOLOv7-plum. (**a**) Example of detection and counting on a Class A image; (**b**) Example of detection and counting on a Class B image; (**c**) Example of detection and counting on a Class C image; (**d**) Example of detection and counting on a Class D image.

**Figure 4 plants-12-02883-f004:**
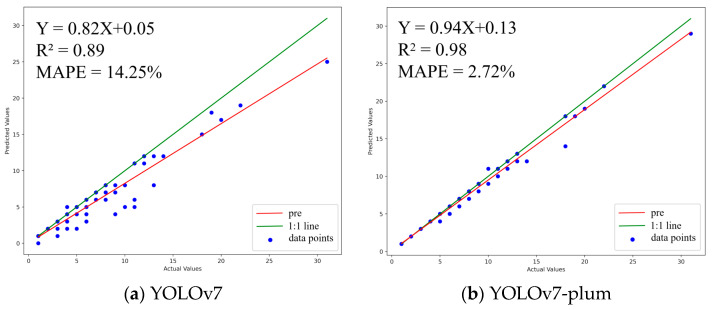
Linear regression plot of YOLOv7-plum and YOLOv7. (**a**) YOLOv7; (**b**) YOLOv7-plum.

**Figure 5 plants-12-02883-f005:**
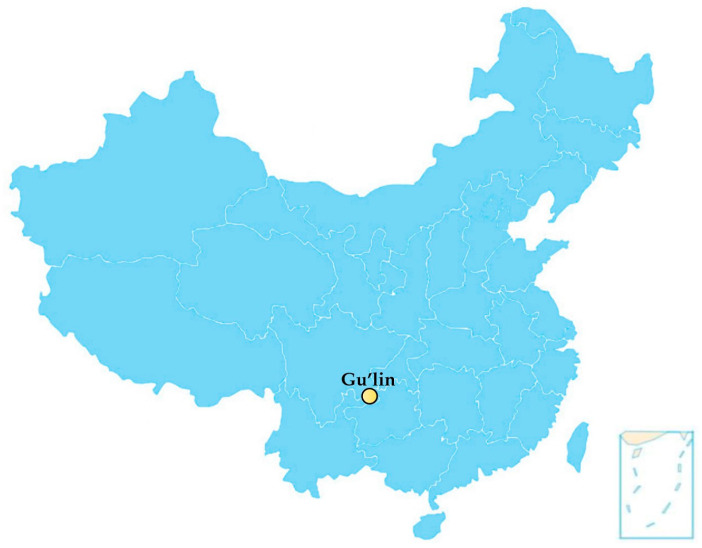
Location of the experimental area in GuLin, as marked on the map.

**Figure 6 plants-12-02883-f006:**
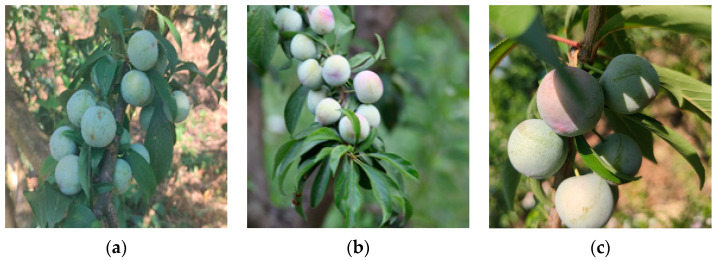
Examples of images taken by different devices. (**a**) Redmi k20 pro; (**b**) Canon 60D; (**c**) SONY Alpha6400.

**Figure 7 plants-12-02883-f007:**
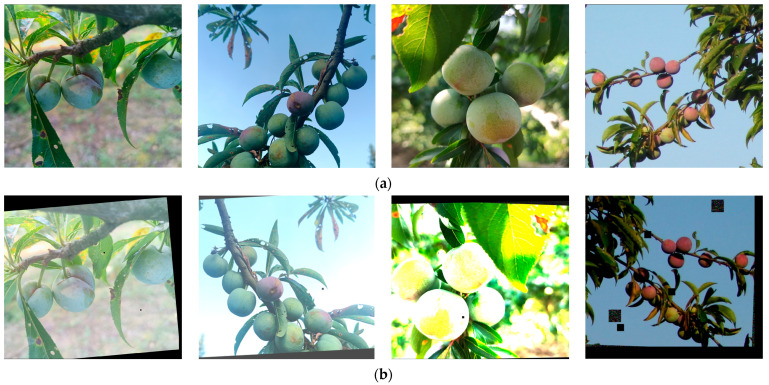
Example of data enhancement. (**a**) Original images; (**b**) Images after data enhancement.

**Figure 8 plants-12-02883-f008:**
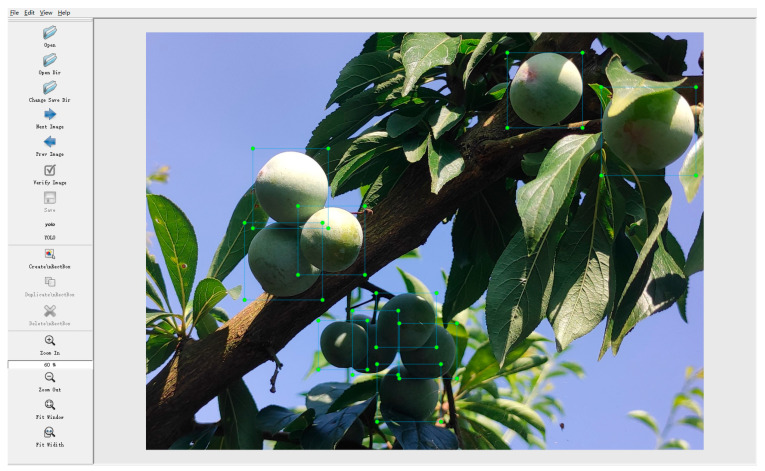
Example of labeling plums.

**Figure 9 plants-12-02883-f009:**
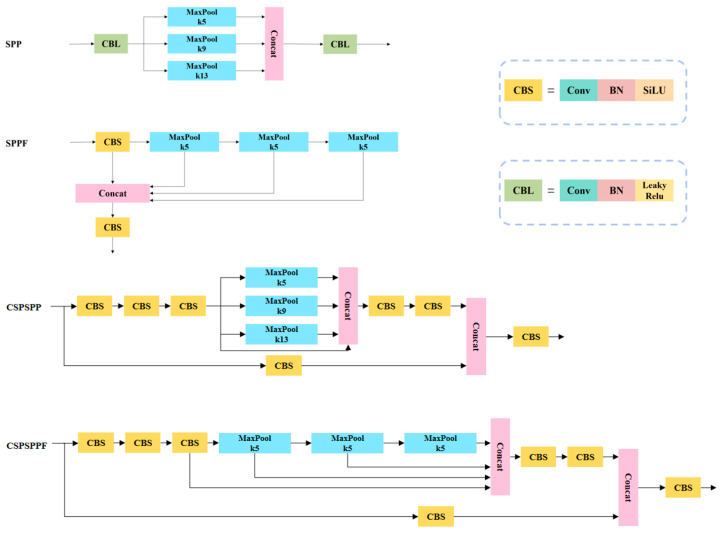
SPP, SPPF, CSPSPP, CSPSPPF structure.

**Figure 10 plants-12-02883-f010:**
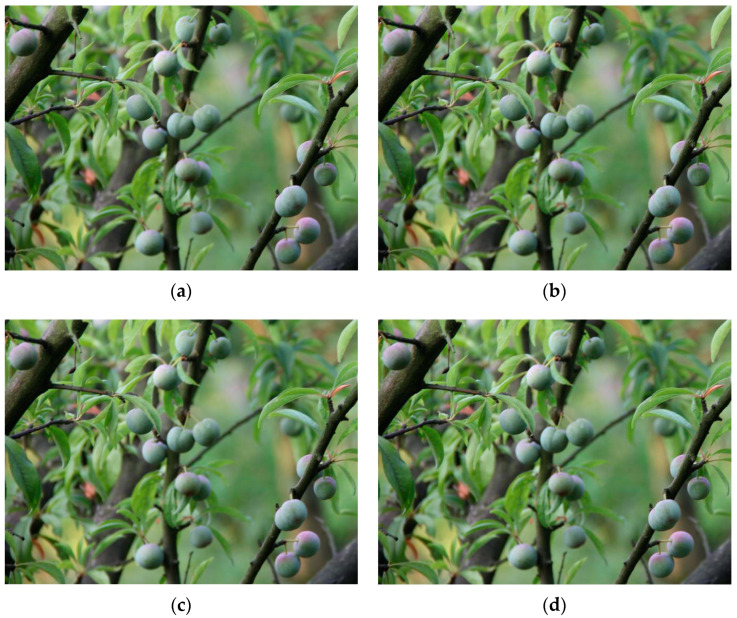
Different interpolation effects. (**a**) original image; (**b**) nearest neighbor interpolation; (**c**) bilinear interpolation; (**d**) bicubic interpolation.

**Figure 11 plants-12-02883-f011:**
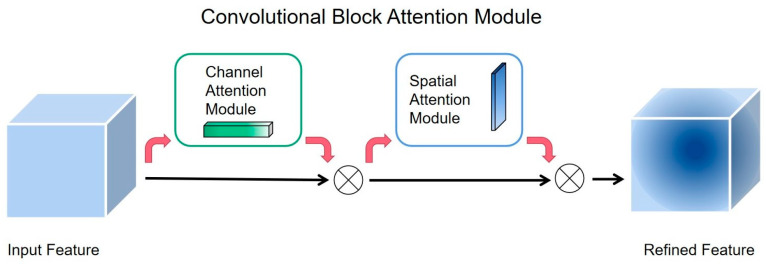
CBAM structure diagram.

**Figure 12 plants-12-02883-f012:**
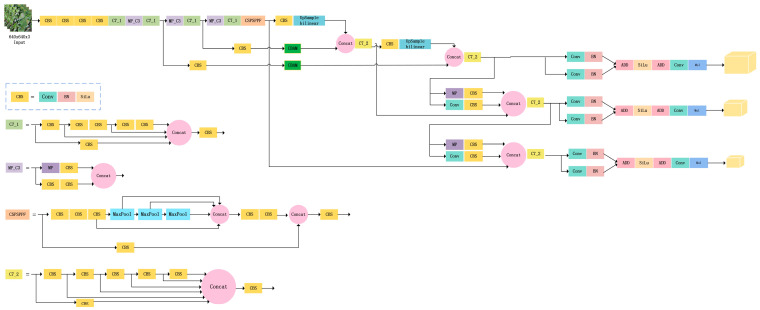
YOLOv7-plum Network structure diagram.

**Table 1 plants-12-02883-t001:** Experimental results of YOLOv7 and YOLOv7-plum on different datasets.

Category	Dataset	Precision	Recall	AP	Model Size
YOLOv7	dataset1	0.9041	0.9079	0.9115	71.3 MB
YOLOv7-plum	dataset1	0.9018	0.8989	0.9233	71.4 MB
YOLOv7	Dataset2	0.9397	0.9239	0.9288	71.3 MB
YOLOv7-plum	Dataset2	0.9342	0.9326	0.9491	71.4 MB

**Table 2 plants-12-02883-t002:** Ablation experiment result.

Model	CSPSPPF	CBAM	Bilinear	Precision	Recall	AP
YOLOv7	×	×	×	0.9397	0.9239	0.9288
YOLOv7-S	√	×	×	0.9389	0.9176	0.9313
YOLOv7-C	×	√	×	0.9264	0.9283	0.9396
YOLOv7-B	×	×	√	0.9342	0.9112	0.9345
YOLOv7-SC	√	√	×	0.9116	0.9373	0.9376
YOLOv7-SB	√	×	√	0.9412	0.9087	0.9394
YOLOv7-CB	×	√	√	0.9236	0.9294	0.9413
YOLOv7-SCB	√	√	√	0.9342	0.9326	0.9491

**Table 3 plants-12-02883-t003:** Comparison results with mainstream models.

Model	Faster R-CNN	YOLOv5	DETR	YOLOv7	YOLOv8l	YOLOv7-Plum
**AP (%)**	90.83	92.68	93.94	92.88	93.90	94.91
**Model Size (MB)**	102	14.4	474	71.3	83.6	71.4

**Table 4 plants-12-02883-t004:** Division of the dataset and distribution of labeled boxes.

Dataset	Percentage	Number of Pictures	Number of Labeled Boxes
training set	80%	976	7042
validation set	10%	122	834
test set	10%	122	869
**Total**	100%	1220	8745

**Table 5 plants-12-02883-t005:** Comparison of time spent on different interpolation algorithms.

Interpolation Algorithm	Original Image	Nearest Neighbor Interpolation	Bilinear Interpolation	Bicubic Interpolation
**Image Size (pixels)**	664 × 498	1325 × 996		
**Execution time (seconds)**		0.008976	0.010970	0.015955

**Table 6 plants-12-02883-t006:** Parameter definition.

Confusion Matrix		Predicted Results	
		Positive	Negative
**Actual Results**	Positive	TP	FN
	Negative	FP	TN

## Data Availability

The data in this study are available on request from the corresponding author.

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
