# Peer review of "YOLOv7-Plum: Advancing Plum Fruit Detection in Natural Environments with Deep Learning"

_plants, 2023, doi:10.3390/plants12152883_

Round 1

Reviewer 1 Report

It is an interesting research. However, the following issues need to be considered.

1. The purpose of this research paper is to count yield and detect pests and diseases, so the real-time performance of the proposed method seems less urgent. On the contrary, high accuracy is the most required in this research.

2.The images collected in the experiment are all local images acquired by cameras, while the purpose of this paper research is to count yield. To count the number of fruits, some global images are required. Do you consider how to achieve global image acquisition in practical applications?

3. The state of art should be strengthened. Some latest detection approaches for related to the research, such as: “Apple grading method design and implementation for automatic grader based on improved YOLOv5” and “Apple target recognition method in complex environment based on improved YOLOv4”, may be referred for the author.

4. Why are the images in Figure 3 reversed after data enhancement?

5.Please compare and explain the results obtained by different methods in Figure 6. In additional, where does the result for “Among these three interpolations, bicubic interpolation works better than bilinear interpolation and nearest neighbor interpolation but is slower than nearest neighbor interpolation and bilinear interpolation in terms of computational speed” in 250-252 lines come from?

6. This paper provides offline validation experimental results of algorithm effectiveness. It would be even better if the design, implementation, and practical application experiments of the algorithm could be provided.

Moderate editing of English language required

Reviewer 2 Report

This study proposes a plum fruit detection model based on an improved YOLOv7. Its main contribution consists in achieving a better plum fruit detection in complex backgrounds when compared with the YOLOv7 standard model.

  • The topic is relevant in the field addressing plum fruit detection in complex backgrounds.
  • It uses an improved deep learning model when compared with state-of-the-art results.
  • The methodology is clearly presented
  • The conclusions are consistent with the evidence and arguments presented and address the main objective
  • The document is well supported with references
  • Figures and Tables are presented clearly

A minor correction in line 320 please standardize text “…yolov7-plum”.

I would advise that the manuscript should be accepted for publication in the present state.

Round 2

Reviewer 1 Report

All my concerns have been resolved in the revision. There are no more comments there.

 Minor editing of English language required.